# Illegal Wildlife Trade and Emerging Infectious Diseases: Pervasive Impacts to Species, Ecosystems and Human Health

**DOI:** 10.3390/ani11061821

**Published:** 2021-06-18

**Authors:** Elizabeth R. Rush, Erin Dale, A. Alonso Aguirre

**Affiliations:** Department of Environmental Science and Policy, George Mason University, Fairfax, VA 22030, USA; erush3@gmu.edu (E.R.R.); edale2@gmu.edu (E.D.)

**Keywords:** amphibians, birds, fish, globalization, emerging infectious disease, illegal wildlife trade, mammals, reptiles, surveillance, zoonotic disease

## Abstract

**Simple Summary:**

Wildlife is trafficked illegally across the globe every day. The illegal wildlife trade (IWT) creates opportunity for emerging infectious disease (EID) events to occur. EIDs are a major threat to wildlife, ecosystems, and public health. This study addresses the lack of comprehensive review of pathogens identified in IWT and highlights the expansion of literature on this subject over the past 30 years. We reviewed 82 scientific papers and conference proceedings from 1990 to 2020. Trends in EIDs identified in IWT have significantly increased over the past decade. Cases covered 240 pathogens across all taxa. Approximately 60% of the pathogens identified were zoonotic (transmissible between animals and humans) and threaten public health. Based on our findings, we recommend further research is needed to monitor and prevent the IWT.

**Abstract:**

Emerging infectious disease (EID) events can be traced to anthropogenic factors, including the movement of wildlife through legal and illegal trade. This paper focuses on the link between illegal wildlife trade (IWT) and infectious disease pathogens. A literature review through Web of Science and relevant conference proceedings from 1990 to 2020 resulted in documenting 82 papers and 240 identified pathogen cases. Over 60% of the findings referred to pathogens with known zoonotic potential and five cases directly referenced zoonotic spillover events. The diversity of pathogens by taxa included 44 different pathogens in birds, 47 in mammals, 16 in reptiles, two in amphibians, two in fish, and one in invertebrates. This is the highest diversity of pathogen types in reported literature related to IWT. However, it is likely not a fully representative sample due to needed augmentation of surveillance and monitoring of IWT and more frequent pathogen testing on recovered shipments. The emergence of infectious disease through human globalization has resulted in several pandemics in the last decade including SARS, MERS, avian influenza H1N1,and Ebola. We detailed the growing body of literature on this topic since 2008 and highlight the need to detect, document, and prevent spillovers from high-risk human activities, such as IWT.

## 1. Introduction

An infectious pathogen recently introduced or reintroduced to a population with an increased incidence or range is defined as an emerging infectious disease (EID) [1]. Novel pathogens continue to emerge, maintaining a trend recorded for the past two decades [2,3]. Changing environments, globalization, human and animal movements, and other drivers contribute to the evolution of infectious agents, allowing them into new ecological niches [4]. Recent EID events attributed to anthropogenic factors include the legal and illegal wildlife trade (IWT—the illicit movement or exchange of wildlife), species overexploitation, habitat loss and degradation, exotic species introduction, global toxification and climate change [5,6]. Because IWT poses many threats to public health, wildlife, and the integrity of ecosystems [5,6,7], it is a primary driver of EIDs. IWT opens new disease transmission mechanisms that lead to zoonotic and epizootic outbreaks [6]. This risk increases with the sale and transport of wildlife away from their endemic regions [8]. EIDs threaten economic stability, biodiversity conservation, and environmental security. Over half of known human pathogens are zoonotic [9]. Many recent zoonotic EID outbreaks, including the emergence of highly pathogenic influenza H5N1, swine influenza H1N1, SARS, Ebola, Marburg, and MERS, originated from wildlife [5,10,11]. 

The emergence of SARS, and most recently SARS-CoV-2, the causative agent of COVID-19, emphasizes the significant economic and social impacts of EIDs; for example, significantly reducing transmission of new zoonotic pathogens from wildlife to humans in tropical forests would cost, globally, between US$22.2 and $30.7 billion each year. Non-pandemic EID outbreaks cost the USA US$80 billion from 1997 to 2009 [12]. SARS impacted the world economy an estimated US$40 billion in 2003 [13]. The global economy will likely lose between US$8.1 and $15.8 trillion due to the COVID-19 pandemic in treatment, vaccine development and reactive approaches to control and eradicate the disease; however, global strategies to prevent pandemics may cost US$22–31B yearly [14,15].

As of 28 May 2021, over 169M human cases and 3.5M deaths have been reported globally due to COVID-19 caused by SARS-CoV-2. It is speculated that the pandemic originated in a wet market in Wuhan, China, suggesting this location was key in its transmission to humans; however, the zoonotic source of its transmission has not been determined. The virus is 79.5% similar to SARS-Co-V and 96.3% similar to Bat-CoV-RaTG13 previously detected in horseshoe bats (*Rhinolophus affinis*) from southwest China’s Yunnan Province. As horseshoe bats hibernated at the time COVID-19 appeared in China, there is general consensus that SARS-CoV-2 has an ancestral origin in Bat-CoV-RaTG13. An amplification host favoring reassortment in the receptor-binding domain (RBD) region was necessary to attach to and infect human cells. No evidence has provided a direct link to a bat or other wildlife species; however, this new coronavirus uses the same receptor (ACE2) as the etiologic agent of the SARS outbreak in 2003, primarily spreading through the respiratory tract [16,17,18].

In addition to the downstream detrimental effects that IWT can cause in human health and economics, global biodiversity loss can also be attributed to the overexploitation of wildlife [19]. Wildlife trade demands exploit numerous wild species and contribute to population declines [20,21,22]. An estimated 30% of species in the world are threatened due to trade [23]. This trade hurts populations, not only due to direct losses from traded goods, but IWT also creates opportunities for spillover events of opportunistic EIDs to occur. The mixing of non-endemic species within the wildlife trade, such as wet markets, is a threat for zoonosis emergence [24] because many wildlife species are reservoirs or incidental hosts for many zoonotic pathogens and can create opportunities for host switching—a commonly discussed mechanism of disease emergence [9]. The pathogens documented in the wildlife trade are diverse [25] and many have spilled over to humans. 

Surveillance and monitoring of IWT is superficial at best and greatly unmonitored compared to legal activity [6,24]. The legal wildlife trade presents many cases of pathogen pollution. These legal cases are likely underreported and even less is known about the risks of IWT [26]. In addition to the general lack of surveillance, many EID monitoring efforts are limited by accessibility of confiscated wildlife and diagnostic tests [27]. A study in 2008 highlighted pathogens identified within IWT [7]. That study presented raw data of pathogens isolated from illegally traded wildlife from 1978 to 2004. The review lacked any analyses and was limited to 14 papers but, for the first time, highlighted cases directly associated with IWT. Others have stated the risks of wildlife trade, but have not performed a comprehensive survey of the literature.

In this review, we aim to expand the 2008 study [7] and build a more extensive database. To the authors’ knowledge, there has not been a compiled list of pathogens identified in wildlife trade from scientific studies other than the 2008 review. Our study highlights specific cases to illustrate the health risks associated with IWT and threats to the conservation of species and ecosystems. While this review is not comprehensive, it does highlight the breadth of pathogens and wildlife in the identifications discovered and published. This review uses case studies from published literature and professional conference proceedings from 1990 to 2020. The types of trade activity within the illegal wildlife trade review include pet trade, bushmeat, human consumption, and wet markets. The need for further surveillance of IWT is echoed not only by other papers, but also by the current COVID-19 pandemic. The cases included in this review offer a glimpse into the health risks to humans and other species associated within IWT [4,6]. 

## 2. Materials and Methods

A review of primary literature from the Web of Science (City of Fairfax, VA, USA) compiled a list of pathogens found in illegally traded wild animals or in humans or domestic animals associated with IWT. Searches were conducted between June 2020 and August 2020. The search terms used included (trade or trading or illegal* or smugg*) and (wild* or animal* or fauna) and (disease* or infection* or pathogen* or parasit* or epidem* or epizoo* or epornit*).

Original searches incorporated 2813, recorded between 2009 and 2020. After discussions about comprehensiveness, the time frame was expanded to include the period 2000–2020, with expanded results for analysis consisting of 3665 records. In July 2020, the search was broadened to the period 1990–2020, resulting in 3862 records for analysis. Subsequent searches were added from the database ProQuest Dissertations & Theses Full Text altered the search terms to limit results about economics, and target species (while limiting to title and abstract searches only). A total of 1100 records were analyzed. The search terms used were: (illegal* or smugg*) and (wild* or animal* or fauna) and (disease* or infection* or pathogen* or parasit* or epidem* or epizoo* or epornit*) and (amphibi* OR mammal* OR reptil* OR insect*). For completeness, we performed searches in the Proceedings of the American Association of Zoo Veterinarians, the Wildlife Disease Association, and the International Associations for Aquatic Animal Medicine. 

Three reviewers analyzed each result and recorded pathogen, species, and trade activity for each result. Each entry included the following information: category; pathogen scientific name; pathogen common name; diagnostic test; taxa affected; species affected scientific name; species affected common name; animal condition; trade activity; species origin (country); species origin (city/state/port); location received (country); location received (city/state/port/market); year documented; year published; notes; and bibliographic reference. All graphics were created by importing Excel data into Tableau Desktop 2020.3 (Salesforce, 2020). The research team grouped specific pathogens, species, animal conditions, and trade activity into general categories for aggregation and summary purposes displayed in the graphics.

## 3. Results

This database included a total of 82 papers listing pathogens found in traded wildlife. Generally, the number of papers focused on wet markets, bushmeat, pet trade, and other illegal trade increased per year since 2008 (Figure 1). Other illegal trade refers to illegal trade not originating as bushmeat, pet trade, or from wet markets. The reports included a total of 240 total pathogens: 85 viruses, 87 bacteria, 12 fungi, eight protozoans, 40 endoparasites, and eight ectoparasites (Table 1). When analyzing illegal trade events (a single paper may refer to multiple species or events), excepting for a few occasions, consecutive rise in the number of species affected by uniquely recorded events began in 2004 reaching 15 reports by 2015 as illustrated (Figure 1).

Birds were the most commonly affected and reported taxa in the dataset. The number of species affected by unique events for birds peaked in 2010 (10), mammals in 2012 (9), and reptiles in 2014 (10) (Figure 2). 

The lack of continuity in monitoring and surveillance limits this review’s analysis about the diversity of pathogens found. However, analyzing the number of pathogens collectively, reporting increased over time with over 34 pathogens reported in 2016 (Figure 3). The majority of papers examined pathogens from one category (Table 1). Five papers included multiple pathogen categories [32,44,50,54,63]. 

The type of illegal trade activity was not evenly distributed geographically and hotspots of activity which involved certain taxa varied. Brazil (22), China (14), and the USA (21) received the most species with zoonotic pathogens totaling 50 bird species, 36 mammals, 28 reptiles, nine amphibians and one fish (Figure 4). The single No Taxa Listed refers to an unspecified shipment of bushmeat. The most common documented activity was illegal trade through travel (44), followed by the pet trade (34), wet markets (27) and bushmeat (20). There were 14 events with unspecified locations. The US, France, Gabon, and Cameroon received bushmeat (Figure 4) and the seizures examined were primarily mammals. Wet market events primarily occurred in China involving reptile species (Figure 5). Pet trade occurred in the USA (9) and Brazil (7) with a few instances in Europe, specifically Spain, Italy, and Greece (Figure 4).

Birds (17), followed by reptiles (10) and mammal species (5) were the most commonly traded taxa in the pet trade. Other documented illegal trade consisted of a variety of taxa affected across the globe including North America, South America, and Asia (Figure 4). In terms of the condition of the animals, in 95 of the 126 (75%) events the animals were live when seized. The other cases involved deceased animals or seized wildlife products. 

The USA (43), France (38), Brazil (32), United Arab Emirates (19), and China (14) imported the greatest number of pathogens through IWT according to the evaluated publications (Figure 6). A total of 23 imported pathogens did not list country of importation. Note, stating these as the highest importers only indicates they have recorded the most through published papers. Few studies report testing of illegal shipments (Figure 1) and this may not encompass the full landscape of true pathogen spillover by country, but rather a focus by researchers on certain countries throughout the case study window.

The diversity of pathogens identified by taxa included 44 different pathogens in birds, 47 in mammals, 16 in reptiles, two in amphibians, one in fish, and one in invertebrate. This is the highest diversity of pathogen types in reported literature related to IWT. This is likely not a fully representative sample. Mammals and birds may not necessarily be more ‘at risk’, other taxa may have as many pathogens, but as surveillance and testing is sparse, the other taxa may simply not be tested as frequently. Enhanced screening of seized IWT shipments would provide insight into the representativeness of this review’s findings, though even seized shipments do not illustrate the full risk of IWT, as many shipments pass across borders unnoticed. 

A total of five events documented zoonotic spillover. These were related to a chlamydiosis outbreak affecting 7/15 Belgian custom officers exposed to smuggled parakeets [28] and monkey pox virus introduced in the USA by various African exotic rodents exposing prairie dogs (*Cynomys* spp.) in the pet trade [56]. Naturally acquired simian retrovirus infections were reported in African hunters through the bush meat trade [105]; an outbreak of opisthorchiasis, a liver fluke infection, contracted by a family from illegally imported raw fish from a highly endemic region [107]. More recently, an outbreak of *Salmonella* Oranienburg was reported from 14 states in the USA affecting 26 people infected with small pet turtles illegally purchased from transient street or roadside vendors [109]. 

Table 1 highlights pathogens and cases with zoonotic potential. Instances where a given pathogen and host have been documented as zoonotic in the past were labeled with a Y (Yes). A pathogen and host with no previous documentation of zoonoses were denoted with a N (No). Papers with multiple hosts and pathogens identified had cases where some pathogen and host combinations were known to have zoonotic potential and others not were labeled with Y/N (Yes/No). Over 60% of the entries have known zoonotic potential (Table 1). 

## 4. Discussion

We know emerging infectious diseases (EIDs) tend to be more prevalent in regions with higher human population density, higher land use for agricultural purposes, intensive livestock and poultry production, and perhaps larger biodiversity (specifically more mammal species). Additionally, EIDs appear more common in tropical regions with high levels of biodiversity (hotspots), vast tropical forests and significant habitat destruction [110]. 

The need to effectively prevent pandemics has never been more obvious; the consequences of neglect have never been clearer. The present crisis reveals the costs of globalization as 30–40 million flights moved over 4 billion passengers a year facilitating a massive and continuous movement of millions of humans and animals across the planet during pre-pandemic times. This global movement combined with climate change, has allowed old pathogens to re-emerge and novel zoonoses such as COVID-19 to emerge, spread and threaten life. Destruction of habitats in many parts of the world and suburban sprawl alter natural equilibria and contact with new species and their pathogens. Habitat destruction and new animal interactions are key in the spread of new diseases. For example, releasing domestic ducks into rice fields where wild ducks also feed provide excellent conditions for viral exchange. The same situation is occurring with bats as habitat loss is causing different species to congregate in the same caves facilitating exchange of viral pathogens. Anthropogenic disturbance of tropical forests has shown the emergence of Lassa fever, Ebola virus, Kyasanur forest disease, and Nipah virus, to name a few [111]. In addition, the risk of spillover is clearly linked to human culture, habits, and behavior related to close contact to several animal groups, i.e., wet markets, bushmeat hunting, illegal trade, and exotic species introduction, but through our research, is likely under documented in scientific literature. Globalization and international travel are known factors increasing the pathogen pool and exposure to novel hosts. It is well known that avian influenza viruses emerge in China and SE Asia (Vietnam) and spread to the rest of the world. Wet markets, bushmeat consumption and over harvesting of wild animal species are not only hastening species extinction, but are changing human–wildlife interactions in a way not witnessed based on behavioral and cultural traits ingrained in humans for millennia [112]. 

The IWT is a transnational environmental crime devastating thousands of vertebrate species generating an estimated yearly illicit revenue of US$5–23 billion, placing thousands of wildlife species in at least 120 countries at risk [113]. Corruption in the source countries, along the supply chain and in destination countries contribute to its growth. IWT and associated illicit supply chains involve both overt and covert human behaviors creating new spaces, reservoirs, exposure pathways, and transmission routes for emerging and resurgent pathogens. Humans are also at significant risk by novel close-contact exposure to new species of animals and their pathogens [22].

A total of seven publications (Figure 1) were reported in a single year. This indicates a continued dearth of well-studied information on this topic in scholarly literature. This number is minimal in comparison to the threat of zoonotic disease spillover. The number of publications per year fluctuated year to year, potentially indicating an increase, but reflecting a sporadic and potentially reactive investment in this area of study. Many more publications involving EIDs in the legal trade and legal pet trade networks were identified, but were out of scope as this research focused on IWT.

The high number of bird references (Figure 2) could be attributed to their popularity within the legal and illegal pet trade and thus more available standardized testing methods. Given the small dataset, it is difficult to draw significant conclusions from any single specific taxa or year. 

Our review highlights the USA, Brazil, and France as the top importing countries of IWT. Based on years of research, the emergence of new pathogens is a global problem and it is very difficult to identify exactly a specific country or region. Too many unknown environmental factors are linked to disease emergence to pinpoint specific, key regions of disease emergence. For example, Ebola was first found in the Democratic Republic of Congo (DRC) and Marburg outbreaks have been detected across Southern Africa. Monkey pox in the USA in 2003 was linked to the wildlife trade from Africa. A decade ago, we documented [7] several diseases linked to the illegal wildlife trade affecting a gamut of vertebrate species; however, they primarily originated from tropical countries. 

Improved ability to detect pathogens in trafficked wildlife will be key to have a clear understanding of potential disease threats introduced into importing countries. A surveillance sample management system for enhanced diagnostic efficiency in collaboration with international partners will be essential to link existing surveillance networks [22]. The USAID’s Emerging Pandemic Threats (EPT) program should be continuously increased to support countries in detecting zoonotic pathogens with pandemic potential. This will include improving laboratory capacity to support surveillance and strengthening national and local response capacities. A large education component for populations at-risk should focus on prevention. For example, PREDICT, a project of EPT, strengthened global surveillance and laboratory diagnostic capabilities for detection of viruses with pandemic potential that are zoonotic. This project discovered over one thousand novel viruses, from which roughly 60% originated from Asia, 40% from Africa, and 7% from Latin America. Most of these viruses were isolated from bats (43%), non-human primates (23%), and rodents (14%) [114]. Efforts such as PREDICT enhance our understanding and feed scientific research to better depict the landscape of EIDs from legal and illegal wildlife sources.

Several epidemics and pandemics occurred within that last decade between SARS and COVID-19 including MERS in the Middle East, 2012; Chikungunya in the Americas, 2013; Ebola in West Africa, 2014; Zika in the Americas, 2015; and Ebola in DRC and Uganda, 2018. These are all zoonotic and linked to human activities. If people are still hunting and consuming animals for traditional medicine, or capturing them for the exotic pet trade, does it really help to put a ban only on consumption of wildlife? In other words, how far would a ban of that limited scope reduce risk? Indeed, this is an issue we need to face. Banning wet markets alone will not solve the problem of the illegal wildlife trade. The problem is more deeply ingrained in behaviors on opposite sides of the spectrum. These markets, on the one hand, provide food for subsistence for those in poverty; on the other, “wild food” has become a fad in countries such as China and is now sought by the affluent as a commodity and social status [22,112].

Poverty, deforestation, habitat loss, and human behavior are key factors leading to EIDs; most likely these triggered the COVID-19 outbreak. Local capacity building, integrative research and transdisciplinary collaborations will be the only way to begin untangle these complex issues that may in many cases result in devastation to humanity [115,116]. In our opinion, simply banning these markets will not make them disappear but may cause them to be driven underground. Therefore, efforts must be made to address all elements of the supply chain and the corruption that facilitates it [22,112]. 

Transdisciplinary science teams are uniquely positioned to help enhance understanding about the intersectionality of wildlife trafficking-related risks and zoonotic pathogens to human health, animal health, and ecosystem health. Together, such teams offer critical refocusing on issues such as priority zoonotic pathogens of biosecurity concern and pandemic potential [22,31,32,33], as well as more transparently testing, identifying, and documented global cases.

We are all interconnected and traditional ecological knowledge asserts that biodiversity is good for our health. In addition, prevention is key and less expensive than control—it is proactive, not reactive. We already have global structures in place for soft governance—the One Health Tripartite Agreement between the FAO, the WHO, the OIE, supported by the World Bank—and building local capacity drives impacts forward. In fact, these international bodies with the United Nations Environment Programme (UNEP) established the much needed One Health High-Level Expert Panel to develop a long-term action plan to improve our understanding on how pandemics can be prevented. We can restore ecosystem function through expanded education and policy change.

## 5. Conclusions

The trends documented for IWT by Gómez and Aguirre (2008) have increased considerably over the past decade due to the complex interactions of EIDs across taxa linked to fundamental anthropogenic and environmental drivers, or perhaps due to increased diagnostic testing. The US seizes far more illegal wildlife than represented in this dataset. To ensure safety of public health, there should be a focus on the monitoring and enforcement of illegal wildlife trade as translocation of goods expands with globalization and the opportunity for emerging diseases intensifies. This study assessed the data available and presents a breadth of pathogen pathways across a wide range of scientific literature, but with more comprehensive monitoring, a more complete quantitative landscape of EID risks in illegal wildlife trade could be achieved.

## Figures and Tables

**Figure 1 animals-11-01821-f001:**
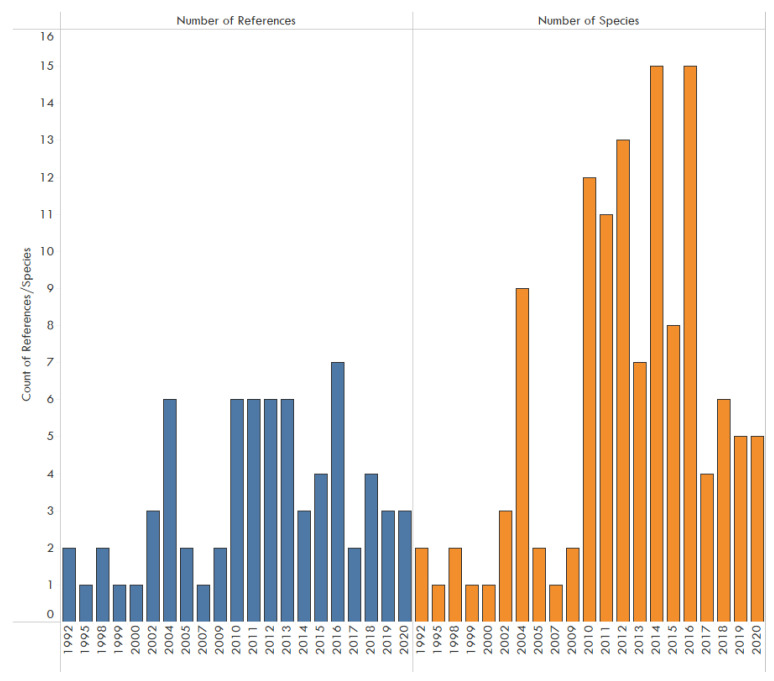
Number of references published and number of species affected by unique illegal trade events (a paper may refer to multiple species), focused on wet markets, bushmeat, pet, and other illegal trade, 1992–2020. Other illegal trade refers to illegal wildlife not originating as bushmeat, pet trade, or wet markets.

**Figure 2 animals-11-01821-f002:**
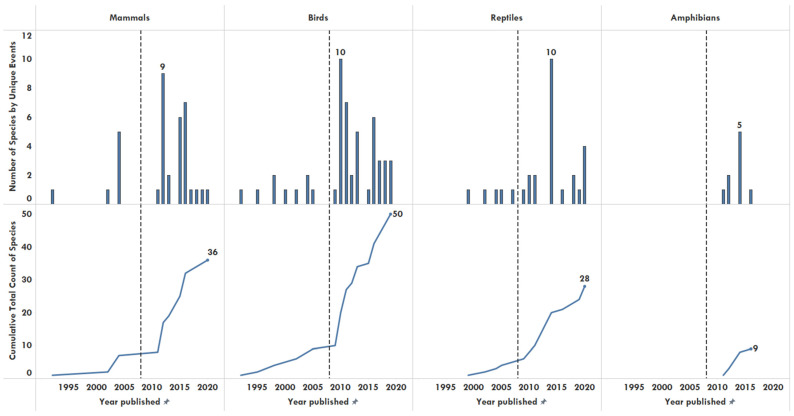
Number of species identified by taxa reported by unique illegal wildlife trade events (top) and cumulative total count of species traded over time, 1990–2020. The reference line indicates the publication of the last review in 2008 [9] in which no amphibian illegal trade was documented.

**Figure 3 animals-11-01821-f003:**
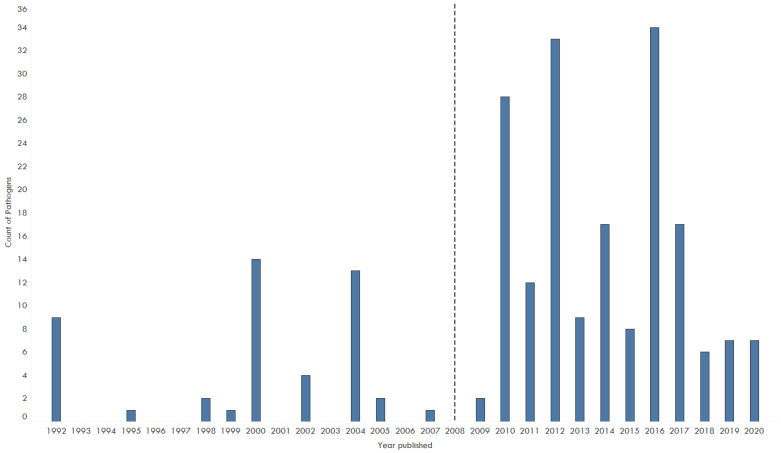
Number of pathogens including viruses, bacteria, endoparasites, ectoparasites and fungi linked to the illegal wildlife trade documented in literature between 1990 and 2020. The reference line indicating the publication of the last review in 2008 [9], in which no amphibian illegal trade was documented.

**Figure 4 animals-11-01821-f004:**
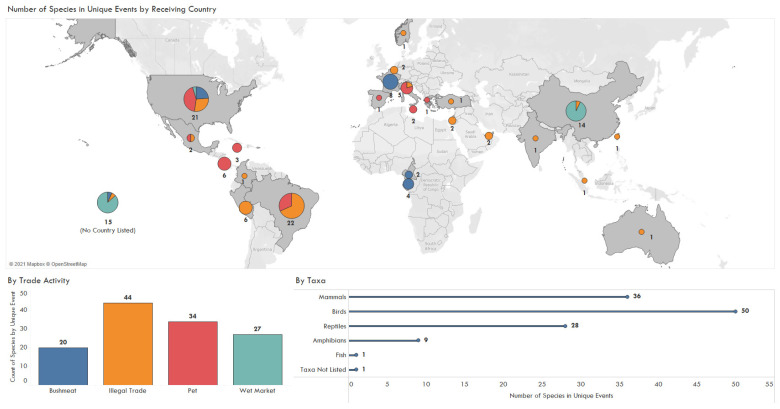
Number of species by taxa and unique trade events including bushmeat, pet trade, wet markets, and other illegal trade, summarized by country of import, 1990–2020.

**Figure 5 animals-11-01821-f005:**
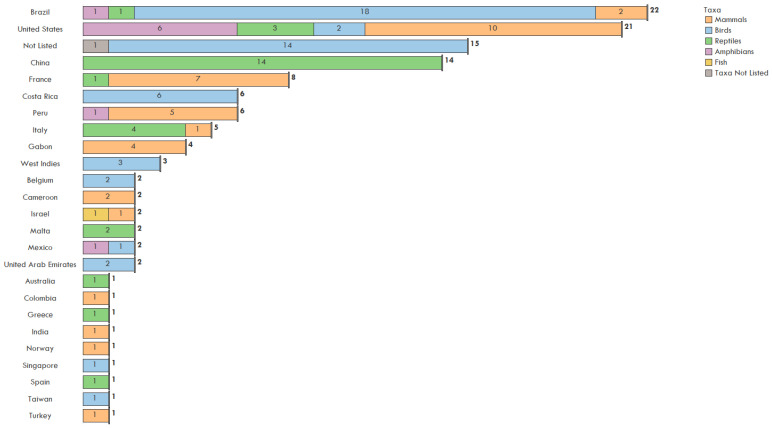
Number of species by taxa including mammals, birds, reptiles, amphibians, and fish from unique illegal wildlife trade events by country of import, 1990–2020.

**Figure 6 animals-11-01821-f006:**
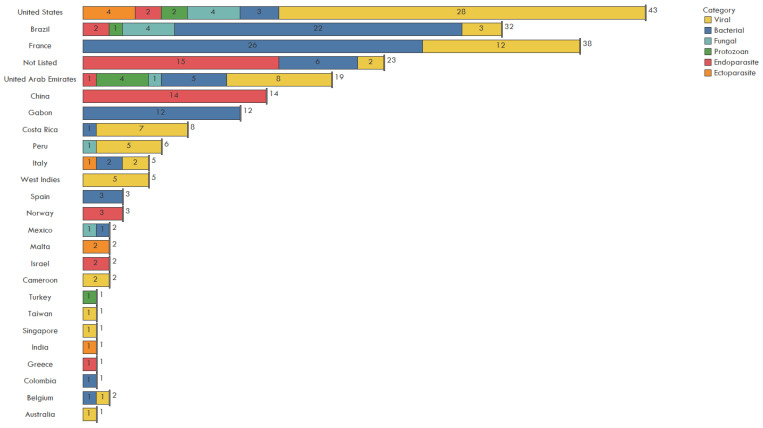
Number of pathogens including viral, bacterial, fungal, protozoan, endoparasites, and ectoparasites introduced through the illegal wildlife trade by country of import, 1990–2020.

**Table 1 animals-11-01821-t001:** Pathogens including viruses, bacteria, fungi, endoparasites, and ectoparasites including genus and species when known, hosts affected, implicated illegal wildlife trade activity and zoonotic potential documented in the literature reviewed, 1990–2020.

Category	Pathogen Category	Affected Category	Species Affected	Trade Activity Category	Zoonotic Potential	References
Viral	Canine coronavirus-II	Mammals	*Canis lupus*	Pet	N	[28]
Bacterial	*Leptospira* spp.	Mammals	*Cebus* sp.	Pet	Y	[29]
Bacterial	*Campylobacter* spp., *Salmonella* spp., *Shigella* spp.	Mammals	N/A	Bushmeat	Y	[30]
Viral	Avian poxvirus, paramyxovirus-1	Birds	*Chlamydotis undulata, C. macqueenii*	Illegal Trade	N	[31]
Bacterial	*Chlamydophila psittaci*, *Clostridium perfringens*, *Pasteurella multocida*, *Pseudomonas aeruginosa*, *Salmonella* spp.	Birds	*Chlamydotis undulata*	Illegal Trade	Y/N	[32]
Endoparasite	Unindentified endoparasites					
Fungal	*Aspergillus fumigatus*					
Protozoan	*Giardia* spp., *Haemoproteus* spp., *Leucocytozoon* spp., *Trichomonas* spp.					
Viral	Adenovirus, avian pneumovirus, avian poxvirus, avian reovirus, paramyxovirus-1, paramyxovirus-2					
Ectoparasite	*Hyalomma aegyptium*	Reptiles	*Testudo graeca*	Illegal Trade	N	[33]
Viral	Newcastle disease virus	Birds	*Amazona oratrix*	Pet	Y	[34]
Bacterial	*Ehrlichia ruminantium*	Invertebrates	*Amblyomma sparsum*	Legal Trade	N	[35]
Fungal	*Batrachochytrium dendrobatidis*	Amphibians	*Telmatobius marmoratus*	Legal Trade	N	[36]
Viral	Avian circovirus	Birds	*Sicalis flaveola*	Illegal Trade	N	[37]
Viral	Avian poxvirus	Birds	Passeriformes	Pet	N	[38]
Bacterial	*Citrobacter freundii*, *Enterobacter* spp., *Klebsiella oxytoca*, *Listeria* spp., *Staphylococcus* spp.	Mammals, Reptiles	N/A	Bushmeat	Y	[39]
Bacterial	*Mycobacterium liflandii*	Amphibians	*Silurana tropicalis*	Legal Trade	N	[40]
Protozoan	Coccidiasina	Birds	*Saltator similis*	Illegal Trade	N	[41]
Ectoparasite	*Amblyomma marmoreum*	Reptiles	N/A	Pet	N	[42]
Endoparasite	*Baylisascaris procyonis*, *Toxascaris leonina*, *Toxocara canis*	Mammals	*Procyon lotor*	Illegal Trade	Y/N	[43]
Ectoparasite	*Cediopsylla simplex*, *Otodectes cynotis*, *Pulex simulans*	Mammals	*Vulpes vulpes*, *Urocyon* *cinereoargenteus*, *Canis latrans*	Illegal Trade	Y/N	[44]
Endoparasite	*Echinococcus multilocularis*, unindentified endoparasites					
Protozoan	*Cystoisospora* spp., *Sarcocystis* spp.					
Viral	rabies					
Bacterial	*Klebsiella pneumoniae*	Birds	Passeriformes, Psittaciformes	Illegal Trade	Y	[45]
Fungal	*Batrachochytrium dendrobatidis*	Amphibians	*Dendrobates tinctorius*	Illegal Trade	N	[46]
Bacterial	*Chlamydophila psittaci*	Birds	budgerigars	Illegal Trade	Y	[47]
Bacterial	*Mycoplasma* spp.	Reptiles	*Cuora bourreti*	Pet	Y	[48]
Viral	Psittacine beak and feather disease virus	Birds	*Ara ambigua*, *Ara macao*	Pet	N	[49]
Fungal	*Batrachochytrium dendrobatidis*	Amphibians	53 different species	Pet	N	[50]
Viral	ranavirus					
Fungal	*Batrachochytrium dendrobatidis*	Amphibians	*Hyla eximia*	Pet	N	[51]
Protozoan	*Toxoplasma gondii*	Mammals	*Bubalus bubalis*	Illegal Trade	Y	[52]
Viral	Simian foamy virus	Mammals	Multiple species	Illegal Trade	Y	[53]
Bacterial	*Citrobacter freundii*, *Enterobacter* spp., *Escherichia coli*, *Klebsiella* spp., *Salmonella* spp.	Birds	Passeriformes	Pet	Y/N	[54]
Endoparasite	Tapeworms, *Trichomonas* spp.					
Fungal	*Aspergillus fumigatus*					
Viral	Avian poxvirus					
Bacterial	*Leptospira* spp.	Mammals	*Saguinus oedipus*	Illegal Trade	Y	[55]
Viral	Monkeypox virus	Mammals	*Cricetomys* sp., *Funiscuirus* sp., *Graphiurus* sp.	Pet	Y	[56]
Fungal	*Aspergillus fumigatus*	Birds	N/A	Illegal Trade	Y	[57]
Viral	H5N1 highly pathogenic avian influenza	Birds	N/A	Illegal Trade	Y	[58]
Viral	Ranavirus	Reptiles	*Chondropython viridis*	Illegal Trade	N	[59]
Ectoparasite	*Amblyomma javanense*	Mammals	*Manis pentadactyla*	Illegal Trade	N	[60]
Viral	Simian foamy virus	Mammals	*Macaca fascicularis*	Legal Trade	Y	[61]
Viral	Morbillivirus	Mammals	*Tragelaphus imberbis*, *Syncerus caffer*, *Aepyceros melampus*, *Taurotragus oryx*	Not Trade Related	N	[62]
Fungal	*Batrachochytrium dendrobatidis*	Amphibians	Multiple species	Illegal Trade	N	[63]
Viral	Ranavirus					
Bacterial	*Chlamydophila* spp., *Escherichia coli*	Reptiles	*Testudo graeca*	Pet	Y	[64]
Endoparasite	*Spirometra* spp.	Reptiles	*Elaphe carinata*, *E. taeniura*, *Zaocys dhumnades*	Wet Market	Y	[65]
Bacterial	*Enterobacter* spp.	Birds	Psittaciformes	Illegal Trade	Y	[66]
Bacterial	*Escherichia coli*	Birds	Psittaciformes	Illegal Trade	Y	[67]
Ectoparasite	*Hyalomma aegyptium*	Reptiles	*Testudo graeca*, *T. kleinmanni*	Pet	N	[68]
Endoparasite	*Ozolaimus megatyphlon*	Reptiles	*Iguana iguana*	Pet	N	[69]
Endoparasite	Acuariidae. Ascaridoidea, Capillarinae, Cestoda, Coccidia	Birds	N/A	Wet Market	Y/N	[70]
Bacterial	*Escherichia coli*	Birds	*Aratinga leucophthalmus*, *Amazonia aestiva*	Pet	Y	[71]
Viral	Herpesvirus	Reptiles	*Testudo kleinmanni*	Pet	N	[72]
Bacterial	*Staphylococcus* spp.	Birds	*Emberizidae* sp, *Thraupidae* sp	Illegal Trade	Y	[73]
Bacterial	*Aeromonas caviae*, *Campylobacter jejuni*, *Salmonella* spp.	Birds	Multiple species	Wet Market	Y	[74]
Viral	Newcastle disease virus	Birds	N/A	Wet Market	Y	[75]
Bacterial	*Leptospira* spp.	Mammals	Multiple species	Illegal Trade	Y	[76]
Viral	Avian circovirus, avian polyomavirus	Birds	*Amazona auropalliata*, *Amazona autumnalis*, *Ara macao*	Pet	N	[77]
Endoparasite	Unindentified endoparasites	Reptiles	Multiple species	Illegal Trade	UNK	[78]
Bacterial	*Escherichia coli*	N/A	N/A	Bushmeat	Y	[79]
Fungal	*Fusarium solani*	Reptiles	*Trachemys dorbigny*	Illegal Trade	Y	[80]
Viral	Human herpesvirus 1	Mammals	*Ateles chamek*	Illegal Trade	Y	[81]
Viral	Aleutian disease virus	Mammals	*Mephitis mephitis*, *Neovison vison*	Legal Trade	Y	[82]
Bacterial	*Chlamydophila psittaci*	Birds	Psittaciformes	Illegal Trade	Y	[83]
Bacterial	*Chlamydophila psittaci*	Birds	*Ara macao*, *Ara ambigua*	Pet	Y	[84]
Viral	Simian immunodeficiency virus	Mammals	N/A	Bushmeat	Y	[85]
Bacterial	*Salmonella* spp.	Reptiles	*Testudo graeca*, *Testudo hermanni*	Pet	Y	[86]
Viral	Herpesvirus	Reptiles	Multiple species	Legal Trade	N	[87]
Bacterial	*Chlamydia psittaci*, *Escherichia coli*, *Shigella* spp.	Birds	Psittaciformes	Pet	Y	[88]
Bacterial	*Chlamydophila psittaci*	Birds	*Anodorhynchus hyacinthinus*	Pet	Y	[89]
Bacterial	*Salmonella panama*, *Salmonella typhimurium*	Birds	*Chrysomus ruficapillus*, *Sporophila falcirostris*	Illegal Trade	Y	[90]
Bacterial	*Escherichia coli*	Birds	Multiple species	Illegal Trade	Y	[91]
Bacterial	*Salmonella* spp.	Reptiles	N/A	Pet	Y	[92]
Fungal	*Batrachochytrium dendrobatidis*	Amphibians	*Lithobates catesbeianus*	Wet Market	N	[93]
Viral	Newcastle disease virus	Birds	N/A	Pet	Y	[94]
Viral	Cytomegalovirus, lymphocryptovirus, simian foamy virus	Mammals	Multiple species	Bushmeat	Y	[95]
Bacterial	*Mycobacterium bovis*	Mammals	N/A	Legal Trade	Y	[96]
Viral	Avian poxvirus, canarypoxvirus	Birds	*Oryzoborus angolensis*, *O. crassirostris*, *Sporophila intermedia*	Pet	N	[97]
Bacterial	*Trepona pallidum*	Mammals	*Gorilla gorilla*	Legal Trade	Y	[98]
Viral	Alphavirus, arenavirus, coronavirus, filovirus, flavivirus, hantavirus, herpesvirus, nairovirus, orthobunyavirus, paramyxovirus, phlebovirus, poxvirus	Mammals	Cercopithecidae	Bushmeat	Y	[99]
Viral	H5N1 highly pathogenic avian influenza	Birds	*Spizaetus nipalensis*	Illegal Trade	Y	[25]
Viral	H3N8 avian influenza	Birds	*Garrulax canorus*	Illegal Trade	N	[100]
Viral	Coxsackievirus	Mammals	*Syncerus caffer*	Not Trade Related	Y	[101]
Bacterial	*Citrobacter* spp., *Enterobacter* spp., *Escherichia coli*, *Hafnia aivei*, *Klebsiella* spp., *Proteus* spp., *Providencia alcalifaciens*, *Salmonella typhimurium*, *Serratia* spp.	Birds	Multiple species	Illegal Trade	Y	[102]
Endoparasite	*Spirometra* spp.	Reptiles	Multiple species	Wet Market	Y	[103]
Viral	Spring viremia of carp virus	Fish	*Cyprinus rubrofuscus*	Legal Trade	N	[104]
Viral	Simian foamy virus	Mammals	N/A	Bushmeat	Y	[105]
Viral	*Trypanosoma lewisi*	Mammals	*Rattus macleari*, *R. nativitatis*	Not Trade Related	N	[106]
Endoparasite	*Opisthorchis* spp.	Fish, Mammals	N/A	Illegal Trade	Y	[107]
Fungal	*Batrachochytrium dendrobatidis*	Amphibians	*Telmatobius* spp.	Illegal Trade	N	[108]

## Data Availability

The data presented in this study are available upon request from the corresponding author.

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
