# Peer review of "Illegal Wildlife Trade and Emerging Infectious Diseases: Pervasive Impacts to Species, Ecosystems and Human Health"

_animals, 2021, doi:10.3390/ani11061821_

Round 1

Reviewer 1 Report

The paper has a good and easy to understand title; the simple summary is also easy to understand without it being simplistic. 

The Abstract is good, but I think it is the authors that detail the growing body of literature rather that the paper (line 29) as the latter is an inanimate object.

The Introduction is well written and provides enough detail for the reader to understand the main issues the authors want to address.

Concerning the methods, these are overall alright, but I am a bit confused about the section starting at line 121. Here diagnostic test is mentioned, species origin and location received (country, city, state, port) but I do not see this back in the Results.

Likewise, graphics are mentioned but these are not included in the paper – I see that these are supposed to be included in the Supplementary Materials but I do not seem to have access to this. Irrespective of this I would included at least a selection of this in the paper itself. The paper is rather short, and in my experience few people consult the supplementary materials. 

The Results are presented very clearly and the table is a very useful resource to have.

Likewise, the Discussion is well written, with sufficient reference to the literature.

The reference list seems complete without missing obvious refernences.

So overall I am impressed with the paper, but unfortunately I have not been able to see the figures – once these are included in the paper I am happy to have another look at it.

Author Response

Abstract i(line 29) as been modified accordingly

Indeed we did not upload figures for this manuscript; however, all figures have been added on this revision solving many of the questions from the reviewers

Reviewer 2 Report

The manuscript is the systematic review of scientific publication concerning cases of illegal wildlife trade and emerging infectious diseases. The article is very interesting and deals with a very current problem. In addition, it shows the need to take steps due to the increasing number of infectious diseases, mainly zoonotic, recorded in illegal wildlife trade. The authors have done a very good job and the article is clearly written. Taking above into account, I recommend publishing the manuscript in “Animals” after introduction  minor corrections (below):

General comment: Figures 1-5 are missing. Please correct the reference numbers in the introduction. The numbers are not consecutive and the references do not always match. Among others, items 2 and 5 are missing, after item 3 appears the number 13. Please read carefully manuscript for some spelling mistakes, e.g. line 54 and 64 should be COVID-19 instead of COVID19 as in line 69. Also font size seems to be different in line 146.  Discussion should be more based on previous literature and more focused on results from review. Some statements are not supported by references.

Detailed comments:

Line 17: How would you introduce prevention of the risk of EIDs? May be it would be better just to write “prevent IWT”?

Line 20: Please correct for “from 1990 to 2020”.

Line 29: Which pandemic do you mean in last decade?

Line 153: please change Leptospira for Leptospira spp.

Line 218-219: please give reference

Line 286-288: Can we say they were all pandemic? Or some of them rather epidemic?

Line 302-303 and 311-319: I agree. However, these are rather personal opinion than scientifically supported statement. I think you should add some reference and in some cases  add “in our opinion”.

Line 321: Is it really so dramatic increase? Don’t you think it may be connected with more testing?

Line 320-322: Conclusion should be much shorten and without reference [116].

Author Response

Figures 1-5 have been added.

Reference numbers have been carefully reviewed and now they are in proper sequence.

Spelling mistakes and fonts have been corrected

Discussion now is supported by references as requested

Detailed comments:

Line 17: edited as requested

Line 20: corrected

Line 29: pandemics have been added

Line 153: Leptospira spp. changed

Line 218-219: reference provided

Line 286-288: Corrected adding  epidemics too

Line 302-303 and 311-319: Fixed as requested adding references too

Line 321: Suggestion added

Line 320-322: Conclusion has been shorten and reference removed

Round 2

Reviewer 1 Report

I have now seen the revised version with the figures included - I think this is a very useful and well evidenced paper. The figures really illustrate well the main findings.

I am happy with the paper as it now is.

Author Response

Thank you for your constructive comments and for no further revisions.

Reviewer 2 Report

The authors revised the manuscript as suggested. After the figures were added, the manuscript became much more understandable and valuable. However,  I still have a few minor comments. After correction will be introduced I believe that the manuscript can be accepted and is of great value. 

General comment: There is no reference in the text to figure 5. Please add it.

Line 149-151: Why did you point out this one? There was also rise in 2004 and 2015 and some significant drops. Maybe this should be mention in discussion with potential explanation. 

Line 156: Figure 1 (caption) - shouldn't it be 1992-2020?

Line 185: Please delate “the”.

Line 221: It seems to me that the reference to the Figure 1 is incorrect at this point. Please change it.

Author Response

 figure 5 is referenced in lines 198-199

Line 149-151: The reference to sudden increase in publications in 2004 has been reworded. We couldn't find an explanation n why those increases and decreases over time

Line 156: years have been fixed to 1992-2020?

Line 185: “the” deleted

Line 221: reference to Figure 1 has been deleted.